# Echinochrome Prevents Sulfide Catabolism-Associated Chronic Heart Failure after Myocardial Infarction in Mice

**DOI:** 10.3390/md21010052

**Published:** 2023-01-12

**Authors:** Xiaokang Tang, Akiyuki Nishimura, Kohei Ariyoshi, Kazuhiro Nishiyama, Yuri Kato, Elena A. Vasileva, Natalia P. Mishchenko, Sergey A. Fedoreyev, Valentin A. Stonik, Hyoung-Kyu Kim, Jin Han, Yasunari Kanda, Keitaro Umezawa, Yasuteru Urano, Takaaki Akaike, Motohiro Nishida

**Affiliations:** 1Division of Cardiocirculatory Signaling, National Institute for Physiological Sciences (NIPS), National Institutes of Natural Sciences, Okazaki 444-8787, Japan; 2Exploratory Research Center on Life and Living Systems (ExCELLS), National Institutes of Natural Sciences, Okazaki 444-8787, Japan; 3Department of Physiological Sciences, SOKENDAI (School of Life Science, The Graduate University for Advanced Studies), Okazaki 444-8787, Japan; 4Department of Physiology, Graduate School of Pharmaceutical Sciences, Kyushu University, Fukuoka 812-8582, Japan; 5G.B. Elyakov Pacific Institute of Bioorganic Chemistry, Far-Eastern Branch of the Russian Academy of Science, Vladivostok 690022, Russia; 6Cardiovascular and Metabolic Disease Center (CMDC), Inje University, Busan 47392, Republic of Korea; 7Division of Pharmacology, National Institute of Health Sciences, Kawasaki 210-9501, Japan; 8Research Team for Mechanism of Aging, Tokyo Metropolitan Institute of Gerontology, Tokyo 173-0015, Japan; 9Graduate School of Pharmaceutical Sciences, The University of Tokyo, Tokyo 113-0033, Japan; 10Graduate School of Medicine, The University of Tokyo, Tokyo 113-0033, Japan; 11Department of Environmental Medicine and Molecular Toxicology, Tohoku University Graduate School of Medicine, Sendai 980-8575, Japan

**Keywords:** echinochrome, myocardial infarction, reactive sulfur species, hydrogen sulfide, sulfide catabolism, cardiac remodeling

## Abstract

Abnormal sulfide catabolism, especially the accumulation of hydrogen sulfide (H_2_S) during hypoxic or inflammatory stresses, is a major cause of redox imbalance-associated cardiac dysfunction. Polyhydroxynaphtoquinone echinochrome A (Ech-A), a natural pigment of marine origin found in the shells and needles of many species of sea urchins, is a potent antioxidant and inhibits acute myocardial ferroptosis after ischemia/reperfusion, but the chronic effect of Ech-A on heart failure is unknown. Reactive sulfur species (RSS), which include catenated sulfur atoms, have been revealed as true biomolecules with high redox reactivity required for intracellular energy metabolism and signal transduction. Here, we report that continuous intraperitoneal administration of Ech-A (2.0 mg/kg/day) prevents RSS catabolism-associated chronic heart failure after myocardial infarction (MI) in mice. Ech-A prevented left ventricular (LV) systolic dysfunction and structural remodeling after MI. Fluorescence imaging revealed that intracellular RSS level was reduced after MI, while H_2_S/HS^−^ level was increased in LV myocardium, which was attenuated by Ech-A. This result indicates that Ech-A suppresses RSS catabolism to H_2_S/HS^−^ in LV myocardium after MI. In addition, Ech-A reduced oxidative stress formation by MI. Ech-A suppressed RSS catabolism caused by hypoxia in neonatal rat cardiomyocytes and human iPS cell-derived cardiomyocytes. Ech-A also suppressed RSS catabolism caused by lipopolysaccharide stimulation in macrophages. Thus, Ech-A has the potential to improve chronic heart failure after MI, in part by preventing sulfide catabolism.

## 1. Introduction

Ischemic heart disease remains the world’s leading cause of death and still shows a tendency to increase in the near future [1]. Myocardial infarction (MI) is the primary clinical manifestation of ischemic heart disease. More than 80% of acute MI is elicited by disruption of atherosclerotic plaque and later thrombosis to induce subsequent occlusion at the coronary artery [2]. Irreversible myocardium ischemia triggers sarcolemmal rupture and causes myocardial cell death [3]. Prolonged loss of myocardium alters loading conditions [4] and increases ventricular wall stress. Consequently, ventricular dilation and myocyte hypertrophy induced by the renin–angiotensin system occur to compensate for the progressive decrease in contractility [5]. Eventually, MI-induced cardiac remodeling will be a predominant cause of arrhythmia, heart failure, and death. Although the survival rate after first infarction has increased to 90%, the long-term prognosis of MI and re-infarction problems are still arduous issues.

Reactive sulfur species (RSS) are defined as redox-active molecules that have catenated sulfur structures, including persulfides (RSSH), polysulfides (RSS(n)SH/RSS(n)SR), and sulfane sulfur. RSS, such as cysteine hydropersulfide (CysSSH) and glutathione persulfide (GSSH), are universally present in living organisms at micromolar levels [6]. Due to the α-effect, the nucleophilicity of adjacent sulfur atom(s) is enhanced [7,8] which renders RSS more active in redox reactions compared to their thiol forms. Therefore, in physiological conditions, RSS can easily reduce or oxidize other molecules [9,10], and thus regulate biological function by Cys modification. RSS such as CysSSH can either enzymatically bond to cysteinyl-tRNA and co-translationally induces protein polysulfidation by incorporating itself into polypeptides or post-translationally modified protein by transpersulfidation [6,11]. Hydrogen sulfide (H_2_S) was previously considered a gaseous transmitter like NO in the last decade [12,13]. However, a recent quantitative study [14] has revealed that cystathionine β-synthase (CBS) and cystathionine γ-lyase (CSE), which were previously committed as major enzymes for synthesizing H_2_S [14,15], was mainly responsible for producing CysSSH. Accumulation of H_2_S anion (H_2_S/HS^−^) inhibits mitochondrial cytochrome c oxidase (COX), which leads to severe respiratory dysfunction with insufficient H_2_S metabolism and excretion [16].

Echinochrome A (Ech-A), or 7-ethyl-2,3,5,6,8-pentahydroxy-1,4-naphthoquinone, is a natural pigment mainly extracted from sea urchins. The naphthoquinone pigments include more than 40 compounds with different pharmacological activities [17] and antiradical activities [18]. Ech-A is heretofore the only commercially available compound that has been applied to medical usage [19], which is also the active ingredient of Histochrome^®^ (Reg. No. in the Russian Federation P N002362/01) [20]. The therapeutic effect of Ech-A has been proven in multiple clinical trials, especially in ophthalmic and cardiac ischemia diseases [20]. Ech-A treatment inhibits the elevation of serum malonic dialdehyde after MI [21]. In clinical studies, Zakirova et al. analyzed the effect of a single intravenous infusion of 100 mg histochrome on MI size and the activity of creatine phosphokinase in 45 patients with acute MI during thrombolytic therapy [22]. Afanas’ev et al. investigated the effect of histochrome on ischemia-triggered ATP depletion in the myocardium of patients with coronary heart disease (CHD). Eight CHD patients received two intravenous injections of 3% histochrome at a dose of 1 mg/kg 24 h prior to operation, and histochrome administration preserved intracellular ATP contents in the myocardium compared with no treatment [23]. In animal models, treatment with Ech-A shows positive outcomes in attenuating cerebral ischemic injury, as well as cardiac ischemia-reperfusion injury [24,25]. The pharmacology potential of Ech-A has been explained by its effective antioxidant [26], anti-inflammatory [27], and anti-fibrotic [28] properties. Ech-A also reportedly protects cardiomyocytes from cardiotoxic agents by alleviating activation of the MAPK signaling pathway and mitochondrial dysfunction [29]. However, the molecular detail of how Ech-A preserves redox homeostasis in the ischemic heart is still unclear.

In this study, we demonstrate whether Ech-A improves chronic heart failure after MI by preventing RSS catabolism to H_2_S. We also demonstrate whether Ech-A inhibits hypoxia-induced RSS catabolism in human iPS cell-derived cardiomyocytes (hiPSC-CMs) and lipopolysaccharide (LPS)-induced RSS catabolism of bone marrow-derived macrophages (BMDMs), suggesting that the mechanism of cardioprotection by Ech-A is through maintenance of RSS metabolism.

## 2. Results

### 2.1. Ech-A Prevents Chronic Heart Failure after MI in Mice

We first examined the chronic effect of Ech-A at 3 doses (0.2, 0.6, and 2.0 mg/kg/day) on left ventricular (LV) dysfunction after MI in mice. We performed echocardiography before and every week after the operation to progressively track the cardiac contraction function and morphology variation (Figure 1B and Table 1). The alleviating effect of Ech-A on the decrease in LV ejection fraction (EF) (Figure 1C) and fractional shortening (FS) (Figure 1D) was gradually manifested with increasing the dose of Ech-A and substantially improved in the MI-Ech-A (2.0 mg/kg/day) group compared to the MI-vehicle group at the end point, indicating that the deterioration of cardiac systolic capacity is dose-dependently suppressed by Ech-A treatment. In addition, LVEF and LVFS have no distinction before infusing with Ech-A in either MI group (Appendix A). The LV anterior wall end diastole (LVAWd) was gradually thinned after MI (Appendix A), while it was greatly restored in Ech-A (2.0 mg/kg/day) treated hearts at 5 weeks after MI (Figure 1E). Furthermore, the elevation of both LV internal diameter end diastole and end systole (LVIDd and LVIDs), as a consequence of LV remodeling, was ameliorated by Ech-A (2.0 mg/kg/day) treatment (Figure 1F,G), as Ech-A treatment tended to attenuate the increase in the ratio of heart weight to tibia length (HW/TL), supporting the prevention of LV remodeling by Ech-A at 2.0 mg/kg/day (Figure 1H and Table 2). The change of each parameter over time can be found in Appendix A. These results suggest that Ech-A administration at 2.0 mg/kg/day (i.p.) improves chronic heart failure after MI.

### 2.2. Ech-A Inhibits LV Remodeling and Oxidative Stress after MI

Since Ech-A at 2.0 mg/kg/day ameliorates chronic heart failure, we next examined the effects of Ech-A on LV remodeling after MI. Additionally, the Ech-A-treated MI heart samples used in all following experiments will be from the 2.0 mg/kg/day group. Picrosirius red staining of histological sections revealed a remarkable fibrosis caused in vehicle-treated MI hearts compared to sham-operated hearts. In contrast, treatment with Ech-A significantly reduced the fibrotic area in the non-infarcted (remote) region of the LV myocardium (Figure 2A,B). Furthermore, significant increases in the myocardial cross-sectional area in the remote region were observed in MI hearts, but they were greatly suppressed by Ech-A treatment (Figure 2C,D). This result correlated well with that of changes in HW (Figure 1H and Table 2) and indicates that Ech-A attenuates LV hypertrophy after MI. Fibrosis occurred in the infarct area, which was mainly due to replacing the necrotic cardiomyocyte with extracellular matrix to form the collagen-based scar. Collagen formation is increased in order to distribute elevated LV wall stress more evenly to stabilize the distending forces and prevent further deformation of the heart [30]. We confirmed that the magnitude of the ischemic scar was equivalently induced in both vehicle-treated and Ech-A-treated MI hearts. In addition, 4-hydroxy-2-nonenal (4-HNE) is a highly cytotoxic and reactive α,β-unsaturated aldehyde that is produced during lipid peroxidation. 4-HNE is a stable marker for oxidative stress and increases in various pathological models, including coronary artery disease [31]. Ech-A treatment inhibited the production of 4-HNE in MI hearts (Figure 2E,F). Ech-A also enhanced the oxidative stress resistance of MI hearts, as indicated by increased mRNA expression levels of *superoxide dismutase 1* and *catalase* (Appendix A). These results suggest that Ech-A treatment improves chronic heart failure after MI by inhibiting oxidative stress and cardiac remodeling.

### 2.3. Ech-A Attenuates RSS Catabolism of the Heart after MI

RSS, such as CysSSH [32] and GSSSG [33], acts as an electron acceptor instead of oxygen, leading to reduced by-production of reactive oxygen species (ROS), while it results in the production of reduced-form sulfides, including H_2_S/HS^−^. We next examined whether Ech-A maintains redox homeostasis by preventing RSS catabolism in MI hearts. SSip-1 DA and SF7-AM probes were applied as R-S_(n)_SH/R-S_(n)_S-R and H_2_S/HS^−^ indicators, respectively [34,35]. Although we could not detect any positive signals of SSip-1 DA and SF7-AM from the scar area of MI hearts, we could semi-quantitatively measure the magnitude of RSS and H_2_S/HS- fluorescence intensities in the non-infarct area. SSip-1 DA imaging revealed that the reduction in RSS in the remote region of MI hearts was restored by Ech-A treatment (Figure 3A,C). Correspondingly, SF7-AM imaging results denoted that Ech-A greatly inhibited the increase in H_2_S/HS^−^ in the non-infarct myocardium region (Figure 3B,D). These results suggest that RSS is catabolized to H_2_S in the LV myocardium after MI, and Ech-A prevents RSS catabolism.

### 2.4. Ech-A Concentration-Dependently Prevents RSS Catabolism Caused by Hypoxic Stress in Neonatal Rat Cardiomyocytes (NRCMs)

We next investigated the relative concentration of RSS and H_2_S/HS^−^ in NRCMs under hypoxic or normoxic conditions. The fluorescence resonance energy transfer (FRET)-based semi-quantitative RSS detection probe (QS10) was used [36]. In parallel to in vivo experimental results, Ech-A treatment significantly suppressed the hypoxic stress-induced decline in intracellular RSS levels in NRCMs (Figure 4A,C). Comparably, hypoxic stress caused an accumulation of H_2_S/HS^−^, which was also suppressed by Ech-A treatment (Figure 4B,D). Furthermore, Ech-A inhibited RSS decrease or H_2_S/HS^−^ increase at a half-maximal inhibitory concentration (IC_50_) of 4.3 μM (Figure 4E) or 2.0 μM (Figure 4F). These results support the in vivo evidence that Ech-A dose-dependently prevents RSS catabolism and H_2_S/HS^−^ accumulation caused by hypoxic stress in cardiomyocytes.

### 2.5. Ech-A Universally Prevents Cellular RSS Catabolism during Hypoxia and Pseudohypoxia

We further investigated whether hypoxia-induced RSS catabolism and H_2_S/HS^−^ accumulation is caused in hiPSC-CMs. Exposure of hiPSC-CMs to hypoxic stress dramatically reduced the intracellular RSS level, and this reduction was attenuated by Ech-A treatment (Figure 5A,B).

The heart contains not only cardiomyocytes but also various cells, including fibroblasts, macrophages, and endothelial cells. Especially, macrophage-derived inflammation is closely related to the progression of cardiac remodeling and heart failure [37,38]. LPS is a major component of Gram-negative bacteria cell walls and induces an inflammatory response in macrophages. Indeed, LPS stimulation decreased intracellular RSS levels in bone marrow-derived macrophages (BMDMs), and Ech-A treatment also suppressed the LPS-induced decline in intracellular RSS levels (Figure 5C,D).

## 3. Discussion

Chronic heart failure due to ischemic heart disease is the world’s leading cause of death. We first demonstrated that treatment with Ech-A one week after MI prevented chronic heart failure in mice, without any apparent side-effects. Although several beneficial effects of Ech-A have been already reported previously, the underlying molecular mechanism(s) is still obscure. Focusing on the antioxidant property of Ech-A, we found that Ech-A has the potential to maintain intracellular redox homeostasis under ischemic or inflammatory stress, by preserving sulfide metabolism using three fluorescent probes (Figure 3, Figure 4 and Figure 5).

During cardiac ischemia or hypoxia, ROS are vastly produced by enzymatic catalyzation [39,40], or reverse the mitochondrial electron transport chain (ETC) [41]. Pathological situations further exacerbate the imbalance between antioxidant systems and ROS, increasing oxidative stress results in membrane protein degradation, DNA damage, and lipid peroxidation. This will lead to autophagosome accumulation, respiratory chain dysfunction, and eventually cause myocardium cell death [42,43]. Antioxidant systems are mainly composed of antioxidant compounds that directly scavenge free radicals and antioxidant enzymes, including superoxide dismutases (SOD) with glutathione peroxidases (GPX) that catabolize ROS. The radical-scavenging property of naphthoquinone pigments relies on their phenolic hydroxyl groups. Since the hydroxyl substituents of naphthoquinone in the C-5 and C-8 (Figure 1A) positions are prone to form hydrogen bonds with quinone carbonyls, hydroxyl groups at C-2, C-3, and C-7 (Figure 1A) are critical for its radical scavenging capacity [17]. Ech-A is hence considered a comparable superb antioxidant. Ech-A has been proven to increase SOD activity [44] and elevate glutathione content with GPX activity [45]. The global antioxidant effect of Ech-A is also reflected in reducing inflammation and tissue damage in the liver [45,46], uvea [47], colon [48], and arteries [49]. These results suggest that not only sulfide metabolism but also the antioxidative effect of Ech-A would contribute to cardioprotective effects.

Hypoxic and inflammatory stress universally reduce intracellular RSS levels in human and rat cardiomyocytes and mouse BMDMs. Consistent with this, brain H_2_S/HS^−^ levels are increased in mice under hypoxic conditions [16]. It has been previously reported that electrons from ETC in mitochondria convert Cys-SSH to H_2_S/HS^−^ [6]. Both oxygen and Cys-SSH are cooperatively used as electron acceptors in ETC. In hypoxic conditions, Cys-SSH would be preferentially used instead of oxygen and converted to H_2_S/HS^−^. Interestingly, Ech-A treatment inhibited the decline in RSS levels after hypoxia. We have no obvious evidence to show how Ech-A inhibits the hypoxic stress-induced RSS catabolism in the present. However, sulfide anabolism and catabolism are precisely controlled enzymatically. For example, mitochondria-localized cysteinyl-tRNA synthetase mediates Cys-SSH formation using L-cysteine as a substrate [6]. CBS and CSE also form Cys-SSH using cystine [14]. Sulfide quinone oxidoreductase (SQOR) catalyzes the oxidation of H_2_S, using glutathione (GSH) as an acceptor to form GSSH. Sulfur dioxygenases such as ETHE1 and persulfide dioxygenase convert GSSH to sulfite, releasing GSH [50]. Chronic Ech-A treatment might change the activity or expression level of these sulfide-metabolizing enzymes. Another possibility is the direct inhibition of Cys-SSH conversion to H_2_S by scavenging electrons. Future studies will be required to identify how Ech-A maintains sulfide metabolism of cells under hypoxic or inflammatory stress.

The relationship between the disturbance of sulfide metabolism and the ischemic vulnerability of the heart is an important open question. Although various physiological roles of externally treated H_2_S/HS^−^ have been reported, H_2_S is generally considered to be a highly toxic gas to most animals. H_2_S induces ROS production and disrupts mitochondrial respiration by inhibiting cytochrome oxidase [51,52]. In mice and rat brains, the expression level of sulfide catabolize enzyme SQOR is significantly lower than that in other tissues, which promotes H_2_S/HS^−^ accumulation [53]. Low expression of SQOR in the brain explains the vulnerability to ischemic (hypoxic) stress by increasing H_2_S production [16]. Although the protective effect of low-dose H_2_S/HS^−^ against chronic heart failure after MI has been reported [54], recent studies have shown that RSS, which is a relatively active metabolite compared to H_2_S, is attracting attention as a key regulator in multiple biological processes [55,56]. Indeed, endogenous H_2_S/HS^−^ level is apparently lower compared to RSS in NRCMs (Figure 3), suggesting that H_2_S/HS^−^ is rapidly converted to RSS in normal cardiomyocytes. Persulfides are chemically highly nucleophilic and redox sensitive, and are therefore comparably more competent at scavenging toxic electrophiles and oxidants than thiol groups [14]. Ech-A treatment protected RSS levels and inhibited oxidative stress in the heart after MI (Figure 2 and Figure 3). Moreover, cysteine persulfide would contribute to energy biogenesis in mitochondrial ETC as an electron acceptor in hypoxia. In addition to low-molecular-weight persulfides, protein persulfidation would also be involved in the ischemic tolerance of the heart. We previously identified that aberrant mitochondrial fission induced by hyperactivation of dynamin-related protein 1 (Drp1) is a mitochondrial fission factor that mediates myocardial senescence and cardiac dysfunction after MI [57]. Drp1 activity is negatively regulated by polysulfidation [6]. Electrophile-mediated depolysulfidation of Drp1 increased cardiac fragility, and NaHS treatment improved mitochondrial and cardiac function by facilitating Drp1 polysulfidation [11]. These results support the beneficial effect of Ech-A on cardiac remodeling and function after MI by protecting the RSS level.

Furthermore, LPS-induced deprivation of RSS in BMDMs and Ech-A treatment improved it (Figure 5C,D). LPS promotes hypoxia-inducible factor 1 (HIF-1) activation in THP-1 human myeloid cells [58], reflecting the induction of pseudohypoxic stress. Consistent with this, LPS induced *hif-1* with its downstream *heme oxygenase-1* gene expression in BMDMs (Appendix A). These results suggest that HIF-1-related signaling might be involved in the regulation of sulfide catabolism under hypoxia and pseudohypoxia. Treatment with RSS donors, such as Na_2_S_4_ and NAC polysulfides, reportedly inhibits hypoxia-induced HIF-1 stabilization and upregulation, increases intracellular RSS levels, and inhibits LPS-induced production of pro-inflammatory cytokines such as tumor necrosis factor-α and interferon-β [55,59,60]. Ech-A has a beneficial effect on various inflammation-related diseases [27]. Therefore, prevention of sulfide catabolism may underlie the anti-inflammatory effect of Ech-A.

## 4. Materials and Methods

### 4.1. Reagents and Antibodies

A total concentration of 10 mg/mL or 37.5 mM Echinochrome A (7-ethyl-2,3,5,6,8-pentahydroxy-1,4-naphthoquinone) diluted in 0.9% sodium carbonate with sodium chloride was acquired from G.B. Elyakov Pacific Institute of Bioorganic Chemistry (P N002363/01, Vladivostok, Russia). SSip-1 DA was purchased from Goryo Chemical, Inc. (Sapporo, Japan). SF7-AM was purchased from Cayman Chemical (Ann Arbor, MI, USA). Collagenase type II was purchased from Worthington Biochemical (Lakewood, NJ, USA). Hoechst stain was purchased from Dojindo (Kumamoto, Japan). Anti-4-HNE antibody was purchased from Abcam (ab46545, Cambridge, United Kingdom). CF594 Anti-Rabbit IgG secondary antibody was purchased from Biotium, Inc. (20152, Fremont, CA, USA). hiPSC-CMs (iCell Cardiomyocytes^2^) were purchased from FujiFilm Cellular Dynamics, Inc. (Madison, WI, USA). QS10 was a gift from Drs. Y. Urano and K. Umezawa.

### 4.2. Animals

All protocols using mice and rats were reviewed and approved by the ethics committees at the National Institute for Physiological Sciences and were conducted according to the institutional guidelines concerning the care and handling of experimental animals. 9 to 12-week-old male C57/BL6J mice and Sprague-Dawley (SD) rats were purchased from Japan SLC, Inc. (Shizuoka, Japan). All mice were kept in plastic cages in a climate-controlled animal room with a 12 h light/dark cycle. Mice were treated with vehicle (0.9% sodium chloride) or Ech-A (0.84, 2.5, or 8.4 mg/mL) via constant intraperitoneal infusion with a mini-osmotic pump (Cat No. 2004, Alzet, DURECT Corporation, Cupertino, CA, USA), which gives a sustained delivery of 0.22 μL of liquid medicine per hour. The dosing concentrations of Ech-A in mice were 0.2, 0.6, and 2.0 mg/kg/day, respectively.

### 4.3. MI Surgery and Transthoracic Echocardiography

Myocardial infarction (MI) was artificially induced by a permeant ligation of the left anterior descending (LAD) coronary artery. The LAD artery was ligated 2 to 3 mm distal to the left atrial appendage with an 8-0 silk suture. The intercostal space, pectoralis major, and skin were successively closed with a 5-0 silk suture. A ligation evoking acute myocardial ischemia was regarded as successful when a blanching of the anterior left ventricle distal from the ligature and a notable ST-segment elevation confirmed by echocardiography were observed. Endotracheal intubation was performed prior to thoracotomy to support the mechanical ventilation of the mice. Sham group mice were performed with the same procedure, except for the LAD ligation. MI surgery was performed on 9 to 12-week-old male C57BL/6J mice. All surgical procedures were performed in mice anesthetized with 3.0% isoflurane (Pfizer, New York, NY, USA) mixed with air for 30 s and stable in 2.0% for MI surgery or 1.0% for echocardiography. A mini-osmotic pump filled with vehicle or Ech-A was implanted intraperitoneally into mice 7 days after MI, and sustained dosing was maintained for 4 weeks. Transthoracic echocardiography was performed using the Vevo3100 imaging system (FUJIFILM VisualSonics, Toronto, Canada) before or every week after the operation. An electrocardiogram was simultaneously recorded by the Vevo3100. Left ventricle fractional shortening (LVFS) was acquired from a parasternal short-axis view of motion-mode echocardiography. 4D-mode left ventricle ejection fraction (LVEF) was calculated according to 3D geometry with dynamic motion of the left ventricle myocardium.

### 4.4. Tissue Analysis

At 5 weeks after MI, mouse hearts were removed, washed in PBS to drain out blood, and fixed with 4% PFA in PBS for 12 h at 4 °C. Then, tissues were successively sunk into 10, 20, and 30% sucrose in PBS every 12 h at 4 °C, and then embedded in optimal cutting temperature (O.C.T.) compound (Sakura Finetek, Torrance, CA, USA) and snap-frozen in a bath of 2-methylbutane with crushed dry ice. Heart slices (12 μm thickness) were cut on a cryostat and used for immunofluorescent and polysulfide/hydrogen sulfide staining.

For immunofluorescent staining, after rinsing with PBS, heart slices were permeabilized and blocked with 1% bovine serum albumin and 0.3% Triton X-100 in PBS at room temperature for 1 h. Then, the slices were incubated with primary antibodies against 4-HNE (1:200) overnight at 4 °C. Next, the slices were incubated with CF594 secondary antibody (1:2000) for 1 h at room temperature. Primary and secondary antibodies were diluted in PBS with 0.05% Triton X-100. Samples were washed three times with PBS and then mounted with ProLong Diamond Antifade Mountant (Invitrogen, Eugene, OR, USA). Slides were observed under a BZ-X700 microscope (Keyence, Osaka, Japan).

For polysulfide and sulfide staining, heart slices were stained with or without 10 μM SSip-1 (for detecting polysulfide) in HBSS including 0.1% BSA and 0.01% cremophor EL or 5 μM SF7-AM (for detecting hydrogen sulfide) in HBSS with 0.04% pluronic F-127 for 45 min at room temperature. Samples were washed three times with PBS and then mounted with ProLong Diamond Antifade Mountant (Invitrogen). Samples were observed using a BZ-X700 microscope (Keyence, Osaka, Japan). For MI samples, only the fluorescent intensity in the non-infarct region of the myocardium was quantified. Background fluorescence intensity without probe was subtracted from fluorescence intensity with probe for quantification.

### 4.5. Cell Culture

Neonatal rat cardiac myocytes were collected from the ventricles of 2-day-old SD rats as previously described [61] then cultured in low glucose (LG) DMEM supplemented with 10% fetal bovine serum (FBS) and 1% penicillin/streptomycin (P/S). Bone marrow-derived macrophages (BMDMs) were prepared as previously described [62]. Briefly, BM cells from mice tibias and femurs were obtained by flushing with DMEM. BM cells were cultured in DMEM supplemented with 20% FBS, L-glutamine, and 30% L929 supernatant containing M-CSF. After 5 days, BMDMs were resuspended in DMEM supplemented with 5% FBS. BMDMs for qPCR were treated with or without LPS (5 ng/mL) and IFNγ (10 ng/mL) for 24 h.

### 4.6. Cell Imaging

NRCMs were seeded at 2.5 × 10^5^ cells/mL on a Matrigel-coated triple-well glass bottom dish (Iwaki) and cultured for 24 h. Then, cells were incubated in LG DMEM (2% FBS, 1% P/S) with 20 μM BrdU for 24 h. Next, cells were preincubated with 10 μM Ech-A in LG DMEM (no FBS, 1% P/S) for 1 h and incubated in 21% or 1% O_2_ condition for an additional 24 h. Last, cells were incubated with 1 μM QS10 (for detecting polysulfide) in HBSS including 2 μg/mL Hoechst and 0.04% pluronic F-127 or 2.5 μM SF7-AM in HBSS including 2 μg/mL Hoechst and 50 μM Mitobright LT (Red) for 30 min at 37 °C. BMDMs were seeded at 2.0 × 10^5^ cells/mL in a four-well glass bottom dish (MATSUNAMI, Osaka, Japan) and cultured for 24 h. DMEM, including 1 or 10 μM Ech-A, was transferred to the plates. After 30 min of incubation at 37 °C, cells were treated with LPS (100 ng/mL) for 7 h at 37 °C to induce the inflammatory response [63]. Cells were incubated with 10 μM SSip-1 DA (for detecting polysulfide) in HBSS, including 0.1% BSA and 0.02% pluronic F-127 for 45 min at 37 °C. Cells were washed twice with HBSS, including 0.1% BSA and 0.02% pluronic F-127. Samples were observed using the BZ-X700 microscope (Keyence, Osaka, Japan). CH1 (Ex: 545 nm, Em: 605 nm) and CH2 (Ex: 545 nm, Em: 700 nm) intensities were measured, and pseudo-color CH1/CH2 ratio images were generated using ImageJ for QS10 analysis. For SSip-1 DA analysis, green fluorescent intensities (Ex: 470 nm, Em: 525 nm) were measured and normalized by red fluorescent intensities (Ex: 545 nm, Em: 605 nm).

### 4.7. RNA Isolation and Quantitative Real-Time Reverse Transcription PCR

For mouse hearts, total RNA from non-infarct heart tissue was purified using the RNeasy Fibrous Tissue Mini Kit (QIAGEN, Hilden, Germany). The RNA concentration and purity were measured by NanoDrop (Thermo Fisher Scientific, Madison, WI, USA). RNA was reverse-transcribed to cDNA using ReverTra Ace qPCR RT Master Mix (Toyobo, Osaka, Japan) according to the manufacturer’s instructions. Quantitative real-time PCR was carried out using KAPA SYBR FAST qPCR Kit Master Mix (2X) Universal (Kapa Biosystems, Wilmington, MA, USA). Reactions were prepared following the manufacturer’s protocol. For BMDMs, total RNA was extracted and cDNA was synthesized as previously described [64]. Quantitative real-time PCR was performed as previously described [64]. Primer sequences used are summarized in Appendix A. To normalize cDNA levels, 18S rRNA expression was used as an endogenous control. Data analysis was performed using LightCycler 96 (Roche, Indianapolis, IN, USA) and Microsoft Excel.

### 4.8. Statistical Analysis

The results are shown as the means ± s.e.m. Statistical comparisons were made with an unpaired t-test (for two groups of variables), an ordinary one-way ANOVA followed by Turkey’s or Šídák’s multiple comparison test (for three and more groups of variables), and a two-way ANOVA followed by Šídák’s multiple comparison test or Dunnett’s multiple comparison test (for three or more groups of variables that change over time). Values of *p* < 0.05 were considered to be statistically significant.

## 5. Conclusions

Chronic administration of Ech-A prevented cardiac dysfunction after MI in a mouse model. Fluorescence imaging analysis showed that hypoxic stress promotes the conversion of intracellular RSS into H_2_S/HS^−^, and this sulfide catabolism is suppressed by Ech-A in rat cardiomyocytes and human iPS cell-derived cardiomyocytes. In this study, we revealed the therapeutic potential of Ech-A for chronic heart failure after MI in mice. Preservation of RSS catabolism and H_2_S/HS^−^ accumulation in cells under hypoxic and inflammatory stress by Ech-A will provide a new strategy for the treatment of chronic heart failure. Future studies focusing on how hypoxic stress induces the imbalance of sulfide catabolism and how Ech-A improves sulfide catabolism are necessary to understand the importance of sulfur metabolism on cardiovascular homeostasis and diseases.

## Figures and Tables

**Figure 1 marinedrugs-21-00052-f001:**
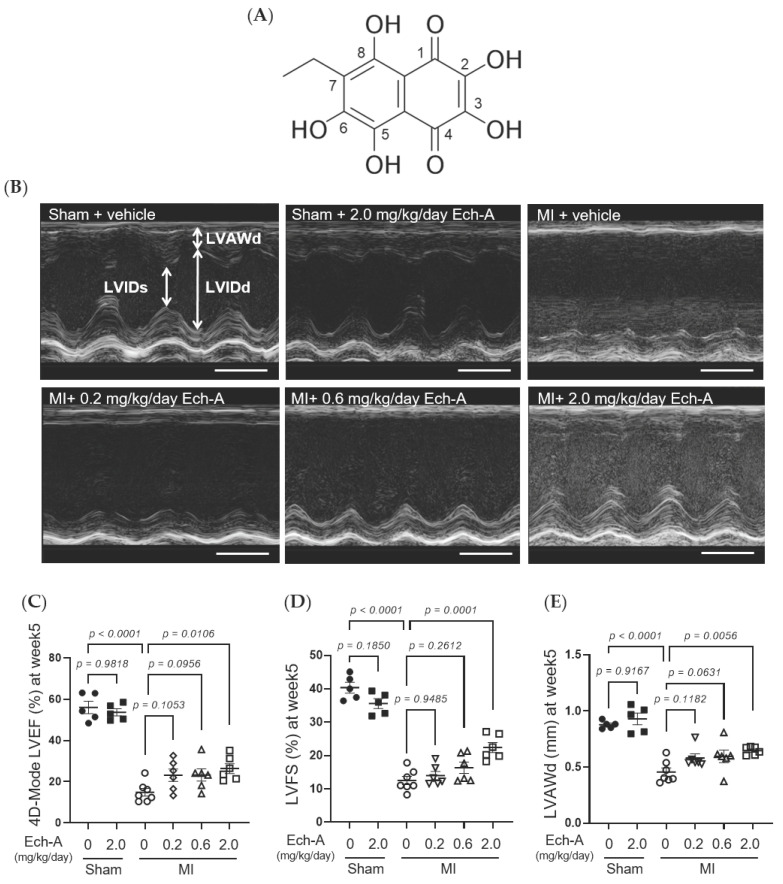
Effects of Ech-A after treatment on MI-induced chronic heart failure in C57BL/6J mice. (**A**) The chemical structure of Echinochrome A, numbering of carbon atoms is represented. (**B**) The representative M-mode echocardiography images of LV 5 weeks in all groups. Scale bar, 0.1 s. (**C**–**G**) Quantification of 4D-mode LV ejection fraction (LVEF) (**C**), LV fractional shortening (LVFS) (**D**), LV anterior wall thickness at end diastole (LVAWd) (**E**), LV internal diameter at end diastole (LVIDd) (**F**) and end systole (LVIDs) (**G**) at end point (5 weeks after MI). Osmotic pump filled with Ech-A or vehicle was implanted intraperitoneally 7 days after MI. (**H**) The heart weight (HW)/tibia length (TL) ratio in mice at 5 weeks after MI (Sham + vehicle (●): *n* = 5, Sham + Ech-A 2.0 mg/kg/day (■): *n* = 5, MI + vehicle (○): *n* = 7, MI + Ech-A 0.2 mg/kg/day (▽): *n* = 6, MI + Ech-A 0.6 mg/kg/day (△): *n* = 6, MI + Ech-A 2.0 mg/kg/day (□): *n* = 6). *p*-values were calculated using one-way ANOVA followed by Šídák’s multiple comparisons test (**C**–**H**).

**Figure 2 marinedrugs-21-00052-f002:**
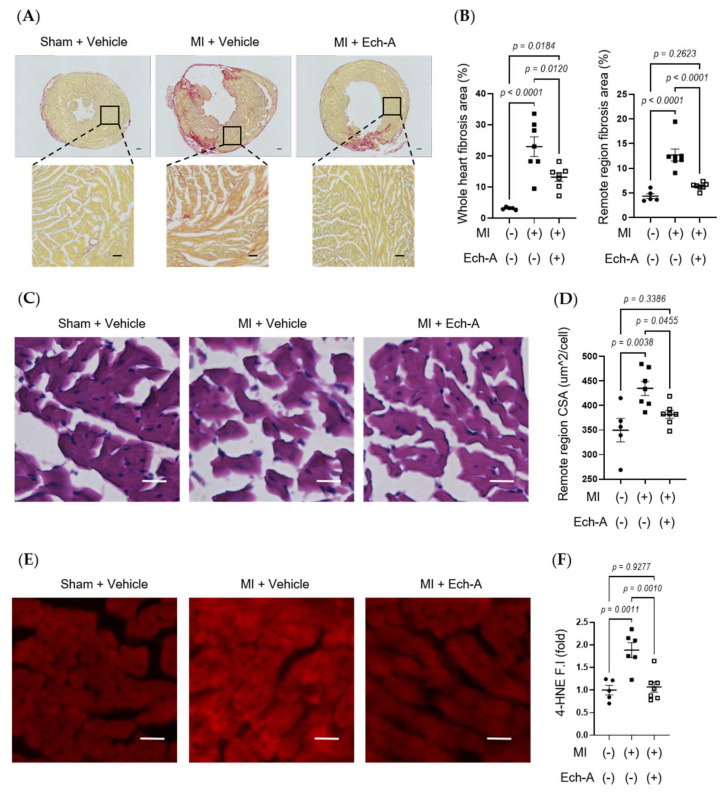
Ech-A treatment suppresses interstitial fibrosis, myocardial hypertrophy, and lipid peroxidation of LV after MI. (**A**) Representative whole heart section and higher magnification images stained with picrosirius red at 5 weeks after MI. Scale bar, 300 μm (**upper**), 100 μm (**lower**). (**B**) Quantitative result of collagen volume fractions in whole hearts and magnified remote areas. (**C**) Representative images of remote regions of LV tissue stained with hematoxylin and eosin at 5 weeks after MI. Scale bar, 30 μm. (**D**) Quantitative result of the cross-sectional area (CSA) of myocardium. (**E**) Representative immunofluorescence images of remote region of LV myocardium at 5 weeks after MI using anti-4-HNE antibody. Scale bar, 30 μm. (**F**) Semi-quantitative result of 4-HNE fluorescence intensity (F.I.). (Sham + vehicle (●): *n* = 5, MI + vehicle (■): *n* = 7, MI + Ech-A (□): *n* = 6). Data were presented as mean ± s.e.m. *p*-values were calculated using one-way ANOVA followed by Turkey’s multiple comparison test.

**Figure 3 marinedrugs-21-00052-f003:**
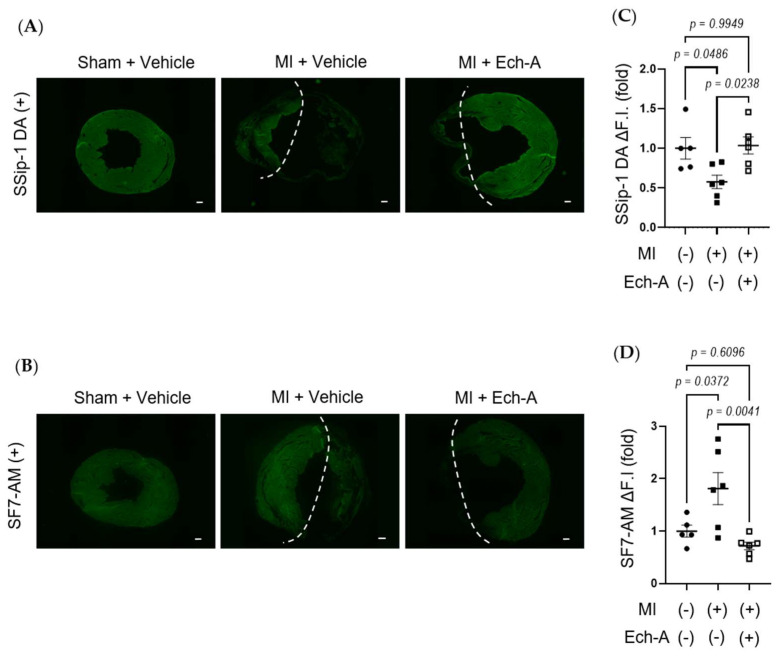
Ech-A treatment prevents RSS catabolism of LV after MI. (**A**,**B**) The representative LV section images of RSS (**A**) and H_2_S/HS^−^; (**B**) levels at 5 weeks after MI. The white dash line separates the infarct area from the non-infarct area of MI hearts. Scale bar, 300 μm. (**C**,**D**) Semi-quantitative results of fluorescent intensities in non-infarct regions of LV (**A**,**B**). (Sham + vehicle (●): *n* = 5, MI + vehicle (■): *n* = 7, MI + Ech-A (□): *n* = 6). Data were presented as mean ± s.e.m. *p*-values were calculated using one-way ANOVA followed by Turkey’s multiple comparison test.

**Figure 4 marinedrugs-21-00052-f004:**
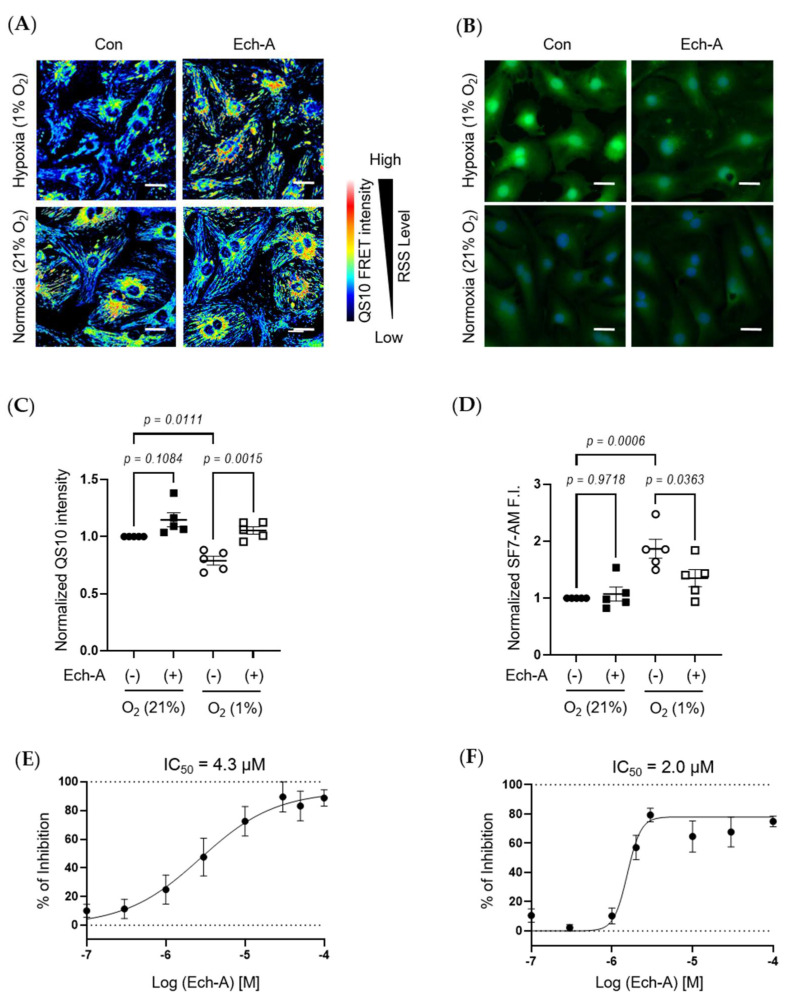
Ech-A treatment concentration-dependently prevents hypoxia-induced RSS catabolism in NRCMs. (**A**,**B**) The representative NRCMs images of RSS (**A**) and H_2_S/HS^−^ (**B**) levels 24 h after normoxia (21% O_2_) or hypoxia (1% O_2_) incubation with or without Ech-A (10 μM). Intracellular RSS or H_2_S/HS^−^ were visualized using QS10 or SF7-AM probe, respectively. Scale bar, 30 μm. (**C**,**D**) Semi-quantitative results of QS10 FRET or SF7-AM fluorescent intensities of (**A**,**B**) (Normoxia + vehicle (●), Normoxia + Ech-A (■), Hypoxia + vehicle (○), Hypoxia + Ech-A (□), *n* = 5 independent experiments). (**E**,**F**) Half-maximal inhibitory concentration (IC_50_) of Ech-A determined by percentage of QS-10 intensity decrease (**E**) and SF7-AM intensity increase (**F**). Average probe intensity at normoxia or hypoxia condition was defined as 100% or 0% grid line, respectively. Data were presented as mean ± s.e.m. *p*-values were calculated using one-way ANOVA followed by Šídák’s multiple comparisons test.

**Figure 5 marinedrugs-21-00052-f005:**
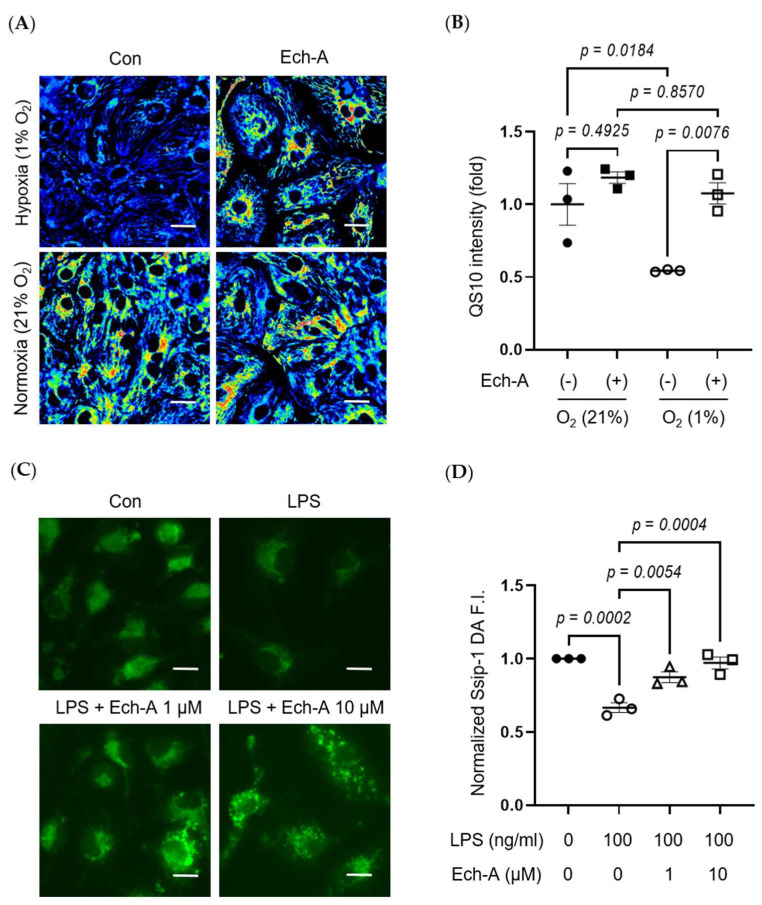
Ech-A suppresses various stress-induced RSS catabolism in vitro. (**A**) Representative hiPSC-CMs images of RSS level after 24 h normoxia (21% O_2_) or hypoxia (1% O_2_) incubation with or without Ech-A concentrations (10 μM). Intracellular RSS were visualized using QS10 probe. Scale bar, 30 μm. (**B**) Quantitative result of QS10 FRET intensities in (**A**) (Normoxia + vehicle (●), Normoxia + Ech-A (■), Hypoxia + vehicle (○), Hypoxia + Ech-A (□), *n* = 3). (**C**) Representative BMDMs images of RSS level 7 h after LPS stimulation with or without different Ech-A concentrations (1 or 10 μM). Intracellular RSS were detected by SSip-1 DA probe. Scale bar, 60 μm. (**D**) Quantitative result of the SSip-1 DA fluorescent intensity in (**C**) (Control + vehicle (●), LPS + vehicle (○), LPS + 1 μM Ech-A (△), LPS + 10 μM Ech-A (□), n = 3 independent experiments). Data were presented as mean ± s.e.m. *p*-values were calculated using one-way ANOVA followed by Šídák’s multiple comparisons test.

**Table 1 marinedrugs-21-00052-t001:** Echocardiographic parameters.

	Sham + Vehicle (Saline)	Sham + Ech-A (2.0 mg/kg/day)	MI + Vehicle (Saline)	MI + Ech-A (0.2 mg/kg/day)	MI + Ech-A (0.6 mg/kg/day)	MI + Ech A (2.0 mg/kg/day)
n	5	5	7	6	6	6
EDV (µL)						
Week 0	41 ± 3	41 ± 3	39 ± 2	40 ± 3	42 ± 2	35 ± 2
Week 1	37 ± 2	36 ± 3	98 ± 15 ∗	76 ± 11	80 ± 12	93 ± 15 ∗
Week 2	38 ± 3	35 ± 3	119 ± 21 ∗	83 ± 15	100 ± 19	89 ± 11 ∗
Week 3	35 ± 1	37 ± 4	127 ± 21 ∗	93 ± 19	99 ± 19	96 ± 13 ∗
Week 4	39 ± 3	35 ± 4	140 ± 27 ∗	88 ± 18	99 ± 20	96 ± 10 ∗
Week 5	37 ± 2	38 ± 2	149 ± 26 ∗	93 ± 18	117 ± 28	100 ± 10 ∗∗
SV (µL)						
Week 0	22 ± 1	21 ± 2	21 ± 1	21 ± 2	22 ± 2	19 ± 1
Week 1	22 ± 2	19 ± 2	19 ± 2	20 ± 1	18 ± 2	21 ± 2
Week 2	22 ± 2	20 ± 2	23 ± 3	18 ± 1	21 ± 2	21 ± 2
Week 3	19 ± 1	18 ± 2	22 ± 3	20 ± 2	21 ± 2	22 ± 2
Week 4	21 ± 2	18 ± 2	19 ± 1	18 ± 1	20 ± 2	23 ± 2
Week 5	20 ± 2	21 ± 1	20 ± 2	19 ± 2	24 ± 3	25 ± 2
CO (mL/min)						
Week 0	9.9 ± 1.0	9.9 ± 0.9	8.5 ± 0.8	8.9 ± 1.2	9.7 ± 1.1	7.9 ± 0.4
Week 1	10.9 ± 1.7	9.7 ± 1.0	8.8 ± 0.8	10.5 ± 1.2	8.6 ± 0.7	11.2 ± 1.1
Week 2	11.4 ± 1.6	10.4 ± 1.1	10.9 ± 1.7	9.2 ± 1.0	9.7 ± 0.9	10.3 ± 1.1
Week 3	9.9 ± 0.9	10.4 ± 1.1	10.8 ± 1.5	10.5 ± 1.7	10.5 ± 1.0	11.0 ± 0.9
Week 4	10.7 ± 1.4	9.4 ± 0.8	9.4 ± 0.9	9.0 ± 1.0	9.7 ± 0.9	10.9 ± 1.4
Week 5	10.6 ± 1.5	10.9 ± 0.2	9.6 ± 1.5	9.0 ± 1.1	11.7 ± 1.9	12.7 ± 1.2
LVAWd (mm)						
Week 0	0.81 ± 0.02	0.81 ± 0.06	0.76 ± 0.05	0.75 ± 0.03	0.74 ± 0.05	0.72 ± 0.02
Week 1	0.80 ± 0.04	0.79 ± 0.06	0.75 ± 0.06	0.64 ± 0.02	0.70 ± 0.02	0.66 ± 0.04
Week 2	0.88 ± 0.07	0.93 ± 0.04	0.57 ± 0.04 ∗	0.64 ± 0.03	0.68 ± 0.03	0.72 ± 0.12
Week 3	0.95 ± 0.05	0.87 ± 0.02	0.54 ± 0.04 ∗∗	0.64 ± 0.05 ∗∗	0.65 ± 0.03 ∗∗	0.78 ± 0.11
Week 4	0.85 ± 0.07	0.92 ± 0.07	0.54 ± 0.06 ∗	0.57 ± 0.03	0.65 ± 0.03	0.72 ± 0.13
Week 5	0.88 ± 0.02	0.93 ± 0.05	0.46 ± 0.04 ∗∗	0.58 ± 0.04 ∗∗	0.60 ± 0.06 ∗	0.65 ± 0.01 ∗∗##
LVPWd (mm)						
Week 0	0.78 ± 0.04	0.87 ± 0.04	0.79 ± 0.05	0.75 ± 0.03	0.83 ± 0.01	0.89 ± 0.10
Week 1	0.92 ± 0.08	0.82 ± 0.05	0.93 ± 0.05	1.00 ± 0.02	0.99 ± 0.10	0.84 ± 0.06
Week 2	1.06 ± 0.11	0.80 ± 0.04	0.88 ± 0.05	0.86 ± 0.05	0.95 ± 0.06	0.83 ± 0.03
Week 3	0.91 ± 0.09	0.89 ± 0.08	0.83 ± 0.05	0.94 ± 0.05	1.02 ± 0.04	0.87 ± 0.09
Week 4	0.92 ± 0.06	0.93 ± 0.09	0.84 ± 0.06	0.98 ± 0.04	0.95 ± 0.05	0.92 ± 0.08
Week 5	0.98 ± 0.11	1.03 ± 0.06	0.81 ± 0.06	0.97 ± 0.06	0.97 ± 0.04	1.04 ± 0.07
LVIDd (mm)						
Week 0	3.93 ± 0.08	3.96 ± 0.09	3.87 ± 0.13	3.95 ± 0.11	4.06 ± 0.10	3.65 ± 0.15
Week 1	3.65 ± 0.09	3.77 ± 0.12	5.20 ± 0.16 ∗∗	4.88 ± 0.22 ∗	4.78 ± 0.18 ∗	5.35 ± 0.18 ∗∗
Week 2	3.60 ± 0.11	3.64 ± 0.08	5.57 ± 0.20 ∗∗	5.08 ± 0.21 ∗	5.29 ± 0.18 ∗∗	5.19 ± 0.27 ∗
Week 3	3.57 ± 0.11	3.73 ± 0.05	5.85 ± 0.20 ∗∗	5.10 ± 0.28	5.32 ± 0.25 ∗	5.13 ± 0.33
Week 4	3.85 ± 0.16	3.50 ± 0.14	6.02 ± 0.26 ∗∗	5.24 ± 0.32	5.51 ± 0.27 ∗	5.10 ± 0.26
Week 5	3.59 ± 0.08	3.79 ± 0.06	6.10 ± 0.29 ∗∗	5.43 ± 0.28 ∗	5.60 ± 0.33 ∗	5.21 ± 0.21 ∗∗
LVIDs (mm)						
Week 0	2.61 ± 0.08	2.69 ± 0.07	2.46 ± 0.13	2.63 ± 0.10	2.69 ± 0.08	2.35 ± 0.20
Week 1	2.43 ± 0.07	2.52 ± 0.12	4.32 ± 0.20 ∗∗	3.93 ± 0.24 ∗	3.89 ± 0.16 ∗∗	4.32 ± 0.20 ∗∗
Week 2	2.13 ± 0.07	2.38 ± 0.08	4.69 ± 0.24 ∗∗	4.13 ± 0.23 ∗∗	4.30 ± 0.19 ∗∗	4.18 ± 0.24 ∗∗
Week 3	2.21 ± 0.10	2.47 ± 0.06	4.96 ± 0.24 ∗∗	4.28 ± 0.32 ∗	4.41 ± 0.26 ∗∗	4.12 ± 0.30 ∗
Week 4	2.39 ± 0.15	2.23 ± 0.18	5.23 ± 0.27 ∗∗	4.45 ± 0.35 ∗	4.61 ± 0.30 ∗∗	3.99 ± 0.21 ∗∗
Week 5	2.14 ± 0.10	2.44 ± 0.06	5.36 ± 0.32 ∗∗	4.68 ± 0.30 ∗∗	4.70 ± 0.36 ∗	4.05 ± 0.20 ∗∗

EDV: end diastolic volume; SV: stroke volume; CO: cardiac output; LVAWd: LV anterior wall thickness at end diastole; LVPWd: LV posterior wall thickness at end diastole; LVIDd: LV internal diameter at end diastole; LVIDs: LV internal diameter at end systole. Data are shown as mean ± s.e.m. ∗ *p* < 0.05, ∗∗ *p* < 0.01 vs. sham + vehicle; ## *p* < 0.01 vs. MI + vehicle. Significance was determined using two-way ANOVA followed by Šídák’s multiple comparison test.

**Table 2 marinedrugs-21-00052-t002:** Organ weight parameters.

	Sham + Vehicle (Saline)	Sham + Ech-A (2.0 mg/kg/day)	MI + Vehicle (Saline)	MI + Ech-A (0.2 mg/kg/day)	MI + Ech-A (0.6 mg/kg/day)	MI + Ech A (2.0 mg/kg/day)
n	5	5	7	6	6	6
BW (g)	27.4 ± 0.3	27.0 ± 0.4	26.9 ± 0.5	26.6 ± 0.5	26.6 ± 0.6	27.0 ± 0.6
HW (mg)	125 ± 4	126 ± 2	192 ± 16 ∗∗	182 ± 8 ∗	189 ± 21 ∗	156 ± 6
HW/BW (mg/g)	4.6 ± 0.2	4.7 ± 0.1	7.1 ± 0.5 ∗∗	6.9 ± 0.3 ∗	7.1 ± 0.8 ∗	5.8 ± 0.2
HW/TL (g/mm)	0.70 ± 0.02	0.68 ± 0.01	1.07 ± 0.09 ∗∗	1.01 ± 0.04 ∗	1.03 ± 0.11 ∗	0.84 ± 0.03
KWl/BW (mg/g)	5.7 ± 0.2	6.6 ± 0.6 ∗	6.0 ± 0.2	6.0 ± 0.1	5.75 ± 0.1	5.9 ± 0.2
KWr/BW (mg/g)	6.0 ± 0.2	6.4 ± 0.2	6.0 ± 0.2	6.3 ± 0.1	6.0 ± 0.1	5.9 ± 0.2
LivW/BW (mg/g)	43.1 ± 1.2	47.6 ± 2.6	42.7 ± 2.6	51.0 ± 0.8	43.7 ± 2.3	44.7 ± 3.3

BW: body weight; HW: heart weight; TL: tibia length; KWl: left kidney weight; KWr: right kidney weight; LivW: liver weight. Data are shown as mean ± s.e.m. ∗ *p* < 0.05, ∗∗ *p* < 0.01 vs. sham + vehicle. Significance was determined using one-way ANOVA followed by Šídák’s multiple comparisons test.

## Data Availability

Not applicable.

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
