# Peer review of "Echinochrome Prevents Sulfide Catabolism-Associated Chronic Heart Failure after Myocardial Infarction in Mice"

_marinedrugs, 2023, doi:10.3390/md21010052_

Round 1

Reviewer 1 Report

Acute myocardial infarction remains a leading cause of morbidity and mortality worldwide,and the incidence is increasing year by year. Although the survival rate after first infarction has raised to 90%, the long-term prognosis of myocardial infarction is still an arduous issue. The manuscript focuses on the prognosis of myocardial infarction, which has certain clinical significance, and the data is presented logically. But there are the following questions for the authors to make some improvement:

Major:

1. The experimental groups are incomplete, whether a “Sham+Ech-A” group should be set up to exclude the effects of Ech-A on the heart ? And based on this, whether the subsequent statistical methods should also be adjusted?

2. Is 4-HNE a specific indicator of oxidative stress? If not, the manuscript said that Ech-A treatment improves chronic heart failure after myocardial infarction by inhibiting oxidative stress, whether a single indicator is too weak to reflect oxidative stress? Maybe other indicators should be added.

3. The manuscript suggests that Ech-A improves chronic heart failure after myocardial infarction by preventing RSS catabolism, is it possible to design experiments to prove the effects of blocking the sulfide metabolism pathway on the prognosis of myocardial infarction to further clarify the conclusion?

4. This manuscript mainly discusses the effects of Ech-A on hypoxia-induced catabolism of RSS, why it suddenly mentioned inflammatory stimulation, what is the relationship between them? And what is the significance of designing this experiment? If it is to prove that inflammatory stimuli can also affect the prognosis of myocardial infarction through sulfide metabolism, is the experiment designed too simple?

5. Do the cells of NRCMs and hiPSC-CMs respond equally to Ech-A? Why hiPSC-CMs were directly treated with a concentration of 10uM?Please give relevant references for the concentration of Ech-A to treat cells.Another related question,please give references for the concentration of LPS to treat BMDMs.

Minor:

1. How to record “a notable ST-segment elevation”? The manuscript doesn't mention the content of electrocardiogram monitoring and the instrument used, is it can be recorded by echocardiography simultaneously?

2. Please clarify the anatomic position of surgical ligation for myocardial infarction,and whether 5-0 silk is too thick for mice?

3. Can you provide representative images of the echocardiography?

4. What does “WT” refer to in Table 1 and Table 2,please note the abbreviation specification and the correspondence between the figure note and the data content.

5. Representative whole heart images can be selected from the same section,the relative position of the scale bar can be adjusted to be consistent.

Reviewer 2 Report

The study is very interesting and relevant, and brings information about Echinochrome-A, which is a very known molecule, it has been used in clinics, but few scientific reports have been published about its effectiveness and safety.

The manuscript is well written and experiments well-chosen and designed. Below some curiosities, doubts, and suggestions:

-          In methods, detail the administration of saline or Ech-A – which volume was injected? It was a constant infusion or in bolus? Why was the mini pump chosen instead of needle?

-          Statistical analysis of table 1 revealed differences to sham+vehicle group. However, I think that differences between MI+vehicle and MI+Ech-A are also important to be checked, although implicit in the previous analysis, to confirm that Ech-A is being active in returning values to the “normal” condition.  The # indicates differences to WT, which is not mentioned in the legend. It should be sham+Vehicle?

-          The antioxidant activity and its relation to the tissue repair should be more explored in the discussion section, in order to explain not only the functional aspects of ‘S’ species removal, but the morphological change that these toxic species cause and what exactly Ech-A do to repair this damaged tissue.

-          Although the focus of the work is sulfur species (and it has been proved according to the shown data), in the discussion section, the participation of ROS in the recovery of cardiac function could be explored. The relationship between oxygen and sulfur species has been commented, but authors could discuss the global antioxidant effect of Ech-A, acting to remove all the reactive species, to have the final effect observed.

Reviewer 3 Report

Xiaokang Tang et al reports about protective effect of Echinochrome on sulfide catabolism-associated chronic heart failure after myocardial infarction in mice..  I have reviewed paper and have next recommendations and questions to the authors:

1.The results in Abstract are too general,. Please indicate real effective concentration and effects by Ech A..

2. Thwe group of naphthoquinone pigments include more that 40 compounds with dufferent pharmacological activitues  (e.g.
https://doi.org/10.1007/s11101-018-9547-3)

3.It is necessary to underline in Introduction, that among other  naphthoquinone pigments only Ech A   has been approved for medicinal use (e.g. https://doi.org/10.1016/j.jep.2019.111933)

4. Antioxidant potential of Ech A was compared with other naphthoquinone pigments  (e.g.
https://doi.org/10.1007/s10337-013-2427-5)

5. The structure of Ech A will be appropriate .

6. Authors have tested only one dose of Ech A on animals. The multidose study is more informative.

7. Figure 1 is too complicated. I suggest to split it. The legend for Fig. 1 require revision after splition of figure. Tested dose of Ech A shoulld be indicate d in the legend.

8. Number of animals should be indicated in the title of tables 1-2. and in the legend s to figures.

9. The resolution of Fig 4B and 5c is low.

10, Please provide batch number for Ech A if it was produced as commercial substance.

11. Please indicate doses of Ech A used in mice and rats.

12. Conclusion section is lacking.

Round 2

Reviewer 1 Report

No further comments for authors.

Reviewer 3 Report

Authors have revised the manuscript according to my recommendations. The manuscript could be accepted in current for,.